# ES-SCLC Patients with PD-L1^+^ CTCs and High Percentages of CD8^+^PD-1^+^T Cells in Circulation Benefit from Front-Line Immunotherapy Treatment

**DOI:** 10.3390/biomedicines12010146

**Published:** 2024-01-10

**Authors:** Anastasia Xagara, Argyro Roumeliotou, Alexandros Kokkalis, Konstantinos Tsapakidis, Dimitris Papakonstantinou, Vassilis Papadopoulos, Ioannis Samaras, Evagelia Chantzara, Galatea Kallergi, Athanasios Kotsakis

**Affiliations:** 1Laboratory of Oncology, Faculty of Medicine, School of Health Sciences, University of Thessaly, GR-41110 Larissa, Greece; xagaraa@hotmail.com (A.X.); alexkokkalis@hotmail.gr (A.K.); tsapakidisk@yahoo.com (K.T.); vasilispapadopoulos1@hotmail.com (V.P.); jnsamaras@gmail.com (I.S.); galkallergi@gmail.com (E.C.); 2Laboratory of Biochemistry/Metastatic Signaling, Section of Genetics, Cell Biology and Development, Department of Biology, University of Patras, GR-26504 Patras, Greece; argyroumi@gmail.com (A.R.); dmtrs.papakonstantinou@gmail.com (D.P.); valiaxantzara@gmail.com (G.K.); 3Department of Medical Oncology, University General Hospital of Larissa, GR-41110 Larissa, Greece

**Keywords:** ES-SCLC, CTCs, PD-L1^+^CTCs, CD8^+^ T-cells, PD-1^+^ T-cells, ICI therapy

## Abstract

SCLC is an aggressive cancer type with high metastatic potential and bad prognosis. CTCs are a valuable source of tumor cells in blood circulation and are among the major contributors to metastasis. In this study we evaluated the number of CTCs that express PD-L1 in treatment-naïve ES-SCLC patients receiving ICI in a front-line setting. Moreover, we explored the percentages of different immune T-cell subsets in circulation to assess their potential role in predicting responses. A total of 43 patients were enrolled—6 of them with LS-SCLC, and 37 with ES-SCLC disease. In addition, PBMCs from 10 healthy donors were used as a control group. Different T-cell subtypes were examined through multicolor FACS analysis and patients’ CTCs were detected using immunofluorescence staining. SCLC patients had higher percentages of PD-1-expressing CD3^+^CD4^+^ and CD3^+^CD8^+^ T-cells, as well as elevated PD-1 protein expression compared to healthy individuals. Additionally, in ES-SCLC patients, a positive correlation between CD3^+^CD8^+^PD-1^+^ T-cells and PD-L1^+^ CTCs was detected. Importantly, patients harboring higher numbers of CD3^+^CD8^+^PD-1^+^ T-cells together with PD-L1^+^CTCs had a survival advantage when receiving front-line immunotherapy. Thus, this study proposes, for first time possible, immune cell–CTCs interaction, as well as a potential novel clinical biomarker for ICI responses in ES-SCLC patients.

## 1. Introduction

Lung cancer is the main cause of cancer-related deaths globally [1]. Small Cell Lung Cancer (SCLC) is a very aggressive sub-type that accounts for 12–15% of all primary lung cancer types [1]. Due to its aggressiveness and high metastatic potential, most of the patients (70–75%) will be diagnosed with extensive stage (ES) disease, leading to poor prognoses [2]. SCLC has a high mutational rate and is also characterized by a lack of actionable, oncogenic driver mutations [3].

Chemotherapy remains the cornerstone of SCLC treatment, with no progress over the last two decades [4]. Recently, the development of immune checkpoint inhibitors (ICIs) has revitalized the hope of improving therapeutic options for SCLC. In 2019, Atezolizumab, a fully humanized antibody against PD-L1, was approved for first-line treatment of ES-SCLC patients in combination with chemotherapy [5]. Following this, an additional anti-PD-L1 ICI antibody, Durvalumab, was also approved in the same setting [6]. However, both approvals are not accompanied by valid predictive biomarkers.

Recently, a molecular subclassification of SCLC patients has been proposed based on the m-RNA expression levels of three transcription factors. SCLC-A expresses high levels of the transcription factor achaete-scute homolog 1 (ASCL1); SCLC-N, high levels of neurogenic differentiation factor 1 (NEUROD1); and SCLC-P expresses POU class 2 homeobox 3 (POU2F3). In addition, the SCLC-I subtype expresses low levels of the mentioned transcription factors and is characterized by an inflamed gene signature [7]. Further analysis of the SCLC-I phenotype revealed high levels of infiltration of different immune cell phenotypes such as T-cells, NK cells, and macrophages, as well as high levels of CD8/PD-L1 expression [8,9]. Subsequently, retrospective analysis of the IMpower133 trial and the CASPIAN trial indicated that the SCLC-I subtype responds to ICI treatment, providing greater OS [8,10]. However, at present, the response to ICI is still limited, and these studies need further validation in prospective cohorts.

Expression of PD-L1 by tumor cells is an immune escape mechanism that reduces the killing capacity of effector T-cells [11]. The evaluation of the PD-L1 protein status of tumor specimens is currently applied as a biomarker for anti-PD-1/PD-L1 therapy [11]. However, there are a lot of limitations in using FFPE tumor samples, such as material availability and the reflection of tumor heterogeneity, which is more pronounced in SCLC where surgery and the biopsy of metastasis is usually not feasible [12]. CTCs are a valuable source of tumor cells in blood circulation and comprise a useful tool for PD-L1 evaluation, and are usually detected in high levels in SCLC patients [13,14].

So far, PD-L1-expressing CTCs have been detected in different types of cancer such as breast, head and neck, and NSCLC [15,16,17,18]. For SCLC, the information about PD-L1+CTCs is limited. Recently, Acheampong and colleagues detected PD-L1-expressing CTCs in the circulation of SCLC patients [19].

In this study, we evaluated the levels of PD-1 expression in circulating T-cells and of PD-L1 in CTCs as a possible mechanism of immune destruction, as well as their possible clinical relevance. To our knowledge, this is the first study regarding ES-SCLC that correlates PD-L1+CTCs and ICs with responses to front-line immunotherapy.

## 2. Materials and Methods

### 2.1. Patients and Blood Collection

A total of 43 chemotherapy-naïve SCLC patients were enrolled in this study. SCLC histological confirmation was performed using patients’ tissue, obtained through bronchoscopy or *EBUS* (endobronchial ultrasound) bronchoscopy. Patients’ median age was 70 years old (range: 44–84), 86% (37/43) of them had extensive disease, and 14% (6/43) had limited stage disease. Most patients (35%) had liver metastasis. All ES-SCLC patients received first-line ICI combined with Carboplatin and Etoposide, while patients with limited stage (LS-SCLC) disease were treated with Carboplatin and Etoposide only. All patients provided written consent for their participation in the study. Study inclusion criteria included: (1) age > 18 years, (2) histologically confirmed diagnosis of SCLC, (3) limited stage or extensive stage disease, (4) being treatment-naïve, (5) general status (Performance Status according to ECOG) 0–2, and (6) written patient consent. Exclusion criteria included: (1) age < 18 years, (2) unconfirmed diagnosis of SCLC, (3) receiving any treatment for cancer (4) general status (Performance Status according to ECOG) PS 0-2, and (5) absence of written informed consent. This was a low-intervention clinical study. Tumor response to therapy was assessed using the Response Evaluation Criteria in Solid Tumors (RECIST 1.1) (Table 1).

From each patient, 30 mL of peripheral blood was collected in K_2_ ethylenediaminetetraacetic acid (EDTA; BD Biosciences, Heidelberg, Germany) at the time of diagnosis and before the administration of any treatment. The study complied with the Ethical Principles for Medical Research Involving Human Subjects according to the World Medical Association Declaration of Helsinki, and was approved by the local ethics and scientific committees of the University General Hospital of Larissa, 41334 Larissa, Greece (32710/3-8-20). All patients provided written informed consent to participate in the study.

### 2.2. Lymphocyte Isolation and Flow Cytometry Analysis

Hypaque-1077 (Sigma-Aldrich, Gillingham, UK) was used to isolate peripheral blood mononuclear cells (PBMCs) of SCLC patients and healthy controls. The isolated cells were re-suspended in RPMI-1640 medium (Biosera, Heathfield, UK), supplemented with 10% heat-inactivated fetal bovine serum (Gibco, Grand Island, NY, USA) and 1% penicillin and streptomycin (Solarbio, Beijing, China). PBMCs were frozen in RPMI-1640 medium supplemented with 20% FBS and 10% DMSO (Sigma-Aldrich, Gillingham, UK) and stored at −80 °C until flow cytometric analysis.

PBMCs were stained for the expression of surface markers using the following anti-human fluorochrome-conjugated monoclonal antibodies: anti-CD3 PE-Cy7; anti-CD4 BV510; anti-CD8 APC-Cy7; anti-CD45RA PE; anti-CD45RO APC; anti-CCR7 FITC; anti-PD-1 PerCPCy5.5; and anti-PD-L1 BV421 (all antibodies were purchased from Biolegend, San Diego, CA, USA). Staining was performed in FACS buffer, 1% PBS-BSA, for 30 min, on ice, in the dark. Acquisition and multicolor analysis were performed using BD FACSChorus Software version 3.0. on a Melody flow cytometer (BD Biosciences, Heidelberg, Germany). For T-cell subsets, the analysis gates were restricted to the lymphocytic population. Each measurement contained 10^6^ single events. Unstained cells were used as negative control, whereas FMO-stained cells were used in order to set the gates.

### 2.3. Isolation and Detection of CTCs

From each of the SCLC patients, 20 mL of peripheral blood was collected and stored in EDTA K2 tubes. The first 5 mL was discarded, to avoid contamination with epithelial cells from the skin. PBMCs were isolated from SCLC patients’ blood using Ficoll-Hypaque (d = 1.077 g/mol) density centrifugation at 1800 rpm for 30 min without breaks. Aliquots of 500,000 cells/500 mL were centrifuged at 2000 rpm for 2 min on Superfrost glass slides (Thermo Fisher Scientific, Waltham, MA, USA).

After triple immunofluorescence staining (CK/PD-L1/CD45), one slide from each patient was analyzed for the identification of CTCs and evaluation of the expression of PD-L1. SCLC patients’ cytospins were fixed and permeabilized with ice-cold aceton:methanol 9:1 (*v*/*v*) for 15 min. Non-specific binding was avoided by blocking with 5% FBS in PBS at 4 °C overnight. Consequently, cells were incubated for 1 h with the anti-CD45 mouse antibody conjugated with Alexa 647 (Santa Cruz Biotechnology, Santa Cruz, CA, USA). The CD45 was used as a negative marker for CTCs. Cytospins were further incubated with goat anti-PD-L1 (Novus Biologicals, Littleton, CO, USA) for 1h, followed by anti-goat Alexa 488 secondary antibody (Life Technologies, Carlsbad, CA, USA) for 45 min. A45-B/B3 mouse antibody was used for the detection of the cytokeratins (CKs) 8/18/19 (Amgen, Southern Oaks, CA, USA), Alexa 555 anti-mouse was used as a secondary antibody (Life Technologies, Carlsbad, CA, USA) for 45 min. Samples were finally mounted on Prolong antifade medium containing DAPI for nuclear visualization. Stained nucleated cells were analyzed using the VyCAP microscopy system version 1.5.1 (VyCAP B.V., Enschede, The Netherlands). Apart from CK-staining, cytomorphological criteria described by Meng et al., 2004 [20] (such as a high nuclear/cytoplasmic ratio and larger cells than white blood cells), were co-estimated in order to characterize a cell as a CTC. Finally, the identification of CTCs was performed blind to clinical data.

### 2.4. Statisical Analysis

Statistical analysis was performed using GraphPad Prism version 6.0 (GraphPad Institute Inc., San Diego, CA, USA). Overall survival (OS) was defined as the time from enrollment in the study until death from any cause, or until the last follow-up that the patient was reported alive. Progression-free survival (PFS) was defined as the time between enrolment in the study and disease relapse, or death, whatever occurred first. Kaplan–Meier analysis was used to correlate immune cell phenotypes, or the presence of CTCs, with patients’ clinical outcomes, with groups being compared using the log-rank test. Differences between groups were determined using the nonparametric Mann–Whitney *U* test. Spearman’s rank correlation tests were used to assess the relationships between the levels of CTCs and tested IC types. For PFS and OS analysis, IC percentages were divided into low and high using the cutoffs defined from receiver operating characteristic (ROC) curves. Cox regression analysis was used to correlate different clinical parameters with survival. Differences and associations were considered significant when *p* < 0.05. All *p* values were two-sided.

## 3. Results

### 3.1. PD-1-Expressing CD8^+^ and CD4^+^ T-Cells in Peripheral Blood of SCLC Patients

The percentages of CD3^+^ T-cells were analyzed in PBMCs through multicolor flow cytometry. Forty-three treatment-naïve SCLC patients were separated into those having limited stage (LS: n = 6) and extensive stage (ES: n = 37) disease. SCLC patients with ES disease had significantly higher percentages of CD3^+^CD8^+^PD-1^+^ (*p* = 0.041) as well as CD3^+^CD4^+^PD-1^+^ (*p* = 0.0029) T-cells compared to HD. Additionally, circulated T effectors (CD45RA^+^CD45RO^−^CCR7^−^) and T effector memory (CD45RA^−^CD45RO^+^CCR7^−^) cells that expressed PD-1 were evaluated between patients and healthy donors. Significantly higher percentages of PD-1-expressing CD8^+^ Teff were detected in ES-SCLC patients (*p* < 0.0001) as well as in LS-SCLC patients (*p* = 0.0003) compared to HD. There were no differences between patients and HD in the percentages of PD-1 of CD4^+^T eff (ES vs. HD: *p* = 0.85; LS vs. HD: *p* = 0.52), CD8^+^ Tem (ES vs. HD: *p* = 0.33; LS vs. HD: *p* = 0.21), or CD4^+^ Tem (ES vs. HD: *p* = 0.19; LS vs. HD: *p* = 0.17) (Figure 1).

### 3.2. PD-1 Expression Levels in SCLC Patients

Quantification of PD-1 expression was determined using median fluorescent intensity (MFI). Both CD3^+^CD8^+^ and CD3^+^CD4^+^ T-cells from LS-SCLC (CD8: *p* = 0.030; CD4: *p* = 0.033) and ES-SCLC (CD8: *p* = 0.0006; CD4: 0.0002) patients were found to express elevated levels of PD-1 compared to HD (Figure 2). Analysis of PD-1 levels on CD8^+^ Teff also indicated elevated expression levels (*p* = 0.0073) compared to HD, but this was not the case for CD4^+^ Teff (*p* = 0.176), CD8^+^ Tem (*p* = 0.0503), or CD4^+^ Tem (*p* = 0.213).

### 3.3. PD-L1-Expressing CTCs and their Correlation with PD-1-Expressing T-Cells

Forty-three SCLC patients were analyzed for CTCs using immunofluorescence staining (Appendix A). Cytokeratin^+^ CTCs (CK^+^) were detected in 42% (18/43) of SCLC patients (Figure 3A). Two of them had LS disease, while sixteen had ES. Phenotypic characterization based on PD-L1 expression indicated that 78% of the CK-positive patients (14/18) had detectable CK^+^CD45^−^PD-L1^+^ CTCs, and 56% (10/18) CK^+^CD45^−^PD-L1^−^ CTCs (Figure 3B). Regarding the average percentages of the total CTCs detected, 60% of them were expressing PD-L1 while 40% were not (Figure 3C, Appendix A).

Correlation analysis was performed between CK^+^CD45^−^, CK^+^CD45^−^PD-L1^−^, or CK^+^CD45^−^PD-L1^+^CTCs and different subtypes of PD-1^+^T-cells. CD3^+^CD8^+^PD-1^+^ T-cells were correlated positively with CK^+^CD45^−^ (Spearman r: 0.351, *p* = 0.038) and CK^+^CD45^−^PD-L1^+^ (Spearman r: 0.342, *p* = 0.044), but not with CK^+^CD45^−^PD-L1^−^ (Spearman r: −0.064, *p* = 0.713) CTCs. There was no correlation between CTCs and the other T-cell populations examined in this study (Table 2). 

### 3.4. Correlation of Circulated T-Cells and CTCs with Clinical Outcome

For each T-cell population, patients with ES-SCLC were dichotomized into high and low using the cutoffs defined by ROC curves. High levels of CD3^+^CD8^+^PD-1^+^ T-cells, at the baseline, were associated with longer PFS (median: 172 vs. 153 days; *p* = 0.037), but not OS (median: 394 vs. 214 days; *p* = 0.065) compared to low levels. Moreover, high levels of CD8^+^ Teff were also associated with higher PFS (median: 180 vs. 158 days; *p* = 0.021), but not OS (median: 261 vs. 232 days; *p* = 0.140) compared to low levels. Additionally, high levels of PD-1^+^CD8^+^ Teff were associated with significantly longer OS (median: 289 vs. 190 days; *p* = 0.008) compared to low levels, although there was no significant difference in PFS (median: 167 vs. 149 days; *p* = 0.191). No other correlation between circulated T-cells with PFS and OS was detected (Table 3).

Regarding the total number of CTCs, no significant correlation to PFS or OS was detected, despite the different examined cutoff values (CTCs > 1 or CK^+^ CTCs > 3) (Appendix A).

As CK^+^CTCs and PD-L1^+^CTCs were found to be associated with CD3^+^CD8^+^PD-1^+^ T-cells, patients were divided into groups as follows: group A—high CD3^+^CD8^+^PD-1^+^ T-cells (roc cutoff) with PD-L1^+^CTCs (n = 13); group B—high CD3^+^CD8^+^PD-1^+^ T-cells (roc cutoff) without PD-L1^+^CTCs (n = 12); group C—low CD3^+^CD8^+^PD-1^+^ T-cells (roc cutoff) with PD-L1^+^CTCs (n = 1); and group D—low CD3^+^CD8^+^PD-1^+^ T-cells (roc cutoff) without PD-L1^+^CTCs (n = 11). Patients in group A were associated with significantly longer OS (*p* = 0.019, med: 394 days vs. 217 days, HR 0.28), but not PFS (*p* = 0.103, med: 193 days vs. 153 days, HR 0.45), compared to patients in group D (Figure 4). No correlation was observed between the other groups or between CD3^+^CD8^+^PD-1^+^ T-cells and CK^+^CTCs (Appendix A).

## 4. Discussion

In the present study, we performed characterization of circulated CD3^+^CD4^+^ and CD3^+^CD8^+^ T-cells in the blood of newly diagnosed ES-SCLC patients treated with immunotherapy on the front line. The results demonstrate high levels of PD-1-expressing CD3^+^CD4^+^ and CD3^+^CD8^+^ T-cells in SCLC patients compared to healthy donors, which has been also reported for other tumor types [21]. Moreover, the association of PD-L1-expressing CTCs with PD-1-expressing T-cell subsets was evaluated. The results demonstrate a positive correlation between circulated PD-L1^+^CTCs and CD3^+^CD8^+^PD-1^+^, but not CD3^+^CD4^+^PD-1^+^ T-cells. This finding is reported for the first time for SCLC patients, indicating an immune surveillance of CTCs. More specifically, in treatment-naïve ES-SCLC patients the percentage of CD3^+^CD8^+^PD-1^+^ T-cells was correlated positively with CTCs expressing PD-L1 in circulation, which is probably linked to the eventual immune evasion of tumor cells. Mechanistically, immune evasion may be performed, at least partly, by the promotion of T-cell exhaustion through the expression of PD-L1 by CTCs. Indeed, CD3^+^CD8^+^PD-1^+^ T-cells from ES-SCLC patients also had reduced levels of protein PD-1 expression compared to healthy controls—a feature of T-cell exhaustion [22].

Immune evasion of CTCs has been also observed by our group in NSCLC patients. High levels of PD-1^+^CD8^+^ T-cells and PD-L1^+^CTCs in circulation were correlated with sorter PFS for the patients who experienced first-line chemotherapy treatment [17]. Moreover, PD-L1+ CTCs were induced after three cycles of chemotherapy, while the numbers of PD-1+ CTCs were consistently reduced and correlated with reduced PFS [23]. Additionally, PD-L1+CTCs were correlated with poorer OS in another cohort of NSCLC patients [24], indicating a possible interaction between CTCs and immune cells in circulation.

Currently, there is no predictive biomarker for ICI in ES-SCLC patients, since most clinical trials are focusing on unselected populations [25]. As a consequence, the outcomes for these patients slightly improved with the addition of ICI to the standard treatment of SCLC [6,26]. In addition, biopsies from metastatic sites are usually not feasible due to the rapid development of the disease, thereby failing to measure the molecular heterogeneity that exists between primary tumors and metastatic lesions [27]. Thus, CTCs’ detection and phenotypic characterization is a promising approach for patient selection.

Indeed, CTCs’ enumeration and analysis can be used in different applications to monitor disease progression and predict responses. It has been shown that CTCs may have a great predictive value in detecting activating mutations, such as epidermal growth factor receptor (EGFR) and EML4-ALK fusion protein, that may be superior to ctDNA analysis [28,29]. We have also shown that live CTCs can be used for functional analysis and evaluation of drug efficacy using TetherChip technology [30].

In 75% of the patients, we detected PD-L1^+^ CTCs, although this was not sufficient to predict responses to ICIs. On the other hand, patients with high levels of CD3^+^CD8^+^ T-cells showed significantly longer PFS and a trend towards better OS, although this was not significant. Additionally, a Cox regression analysis revealed no significant correlation between different clinical parameters—such as age, metastatic site, and smoking status—and patients’ survival (Appendix A) More importantly, patients with PD-L1^+^CTCs isolated from treatment-naïve ES-SCLC and high levels of PD-1^+^-expressing CD3^+^CD8^+^ T-cells have a survival advantage when treated with the standard chemotherapeutic regimens together with ICI in a front-line setting. This finding provides strong evidence for immune destruction by CTCs. It is widely accepted that the induction of PD-L1 by tumor cells is an immunosuppressive mechanism that leads to T-cell exhaustion [31]. Exhausted T-cells express high levels of inhibitory receptors including PD-1 and have low cytotoxic functions [22]. Thus, ICIs block inhibitory signals, thereby re-invigorating T-cells [32].

Circulated PD-1^+^—but not PD-1^−^—T-cells have been also correlated with clinical outcomes in different tumor types, as they reflect neoantigen-reactive T-cells that reside within the tumor [33,34]. Many studies indicate that elevated percentages of CD8+PD-1+ T-cells that usually have an effector phenotype can be used as a non-invasive surrogate biomarker for immunotherapy response in solid tumors such as gastric cancer, RCC, and NSCLC [35,36,37]. Similarly, in this study, we report that in ES-SCLC patients, high levels of circulated PD-1-expressing CD8+ effector cells are correlated with OS benefit when treated with ICI. In contrast, CD4+PD-1+ T-cell subsets that were examined could not be related to patients’ clinical benefit.

Conclusively, this study revealed a strong positive correlation between PD-1 expression on T-cells and PD-L1-expressing CTCs in the circulation of ES-SCLC patients. More importantly, patients with high percentages of both CD3^+^CD8^+^PD-1^+^ T-cells and PD-L1^+^ CTCs had a survival advantage when treated with ICI in a first-line setting. This combination could be proposed as a potential biomarker of ICI response. However, due to the explorative nature of the study, the results should be confirmed in a larger cohort. Moreover, in the future, including a higher number of LS-SCLC patients that are currently undergoing chemotherapy as a front-line treatment may reveal a benefit for ICI administration in a cohort of these patients.

## Figures and Tables

**Figure 1 biomedicines-12-00146-f001:**
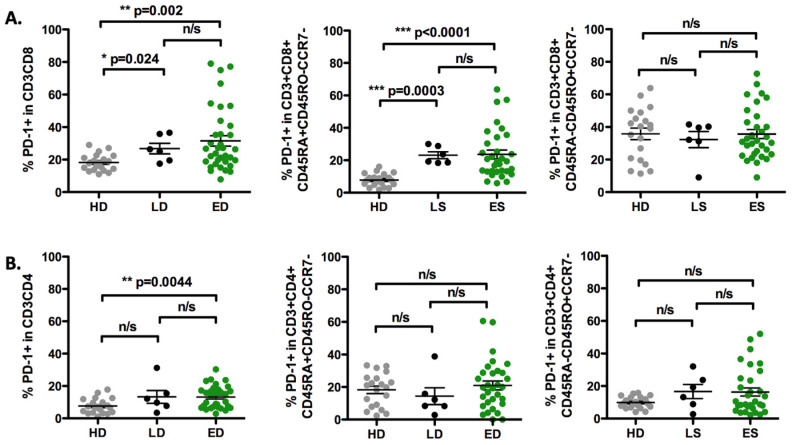
Percentages of PD-1^+^ T-cells in circulation from healthy donors and treatment-naïve SCLC patients. Graphs show frequency of PD-1+ cells among the different subsets described as CD3^+^, CD45RA^+^CD45RO^−^CCR7^−^ (T eff), and CD45RA^−^ CD45RO^+^CCR7^−^ (Tem) on (**A**) CD8^+^ T-cells and (**B**) CD4^+^ T-cells. HD: healthy donors (n = 20); LS: limited stage SCLC (n = 6); ES: extensive stage SCLC (n = 37); n/s: non-significant. Lines represent group sizes.

**Figure 2 biomedicines-12-00146-f002:**
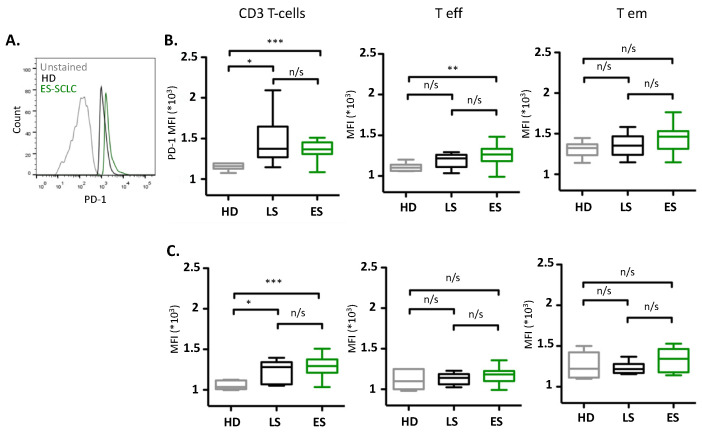
PD-1 MFI on circulated T-cells of healthy donors and treatment-naïve SCLC patients. (**A**) Representative FACS histograms of CD3^+^CD8^+^PD-1^+^ T-cell subsets from HD and ES-SCLC PBMCs. (**B**) MFI values of PD-1 among the different subsets described as CD3^+^, CD45RA^+^CD45RO^−^CCR7^−^ (T eff), and CD45RA^−^ CD45RO^+^CCR7^−^ (Tem) on CD8^+^ T-cells and (**C**) CD4^+^ T-cells. HD: healthy donors (n = 10); LS: limited stage SCLC (n = 6); ES: extensive stage SCLC (n = 37); n/s: non-significant; * *p* < 0.05; ** *p* < 0.001; *** *p* < 0.0001. Unpaired Student *t* test.

**Figure 3 biomedicines-12-00146-f003:**
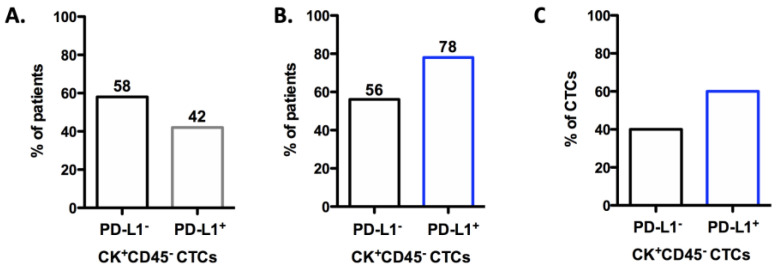
CTCs in treatment-naïve SCLC patients’ blood. (**A**) Percentage of patients with CK+ CD45- CTCs. (**B**) Percentage of patients with CK+CD45-PD-L1- and CK+CD45-PD-L1+ phenotypes. (**C**) Average percentages of the total CTCs detected with CK+CD45-PD-L1- and CK+CD45-PD-L1+ phenotypes.

**Figure 4 biomedicines-12-00146-f004:**
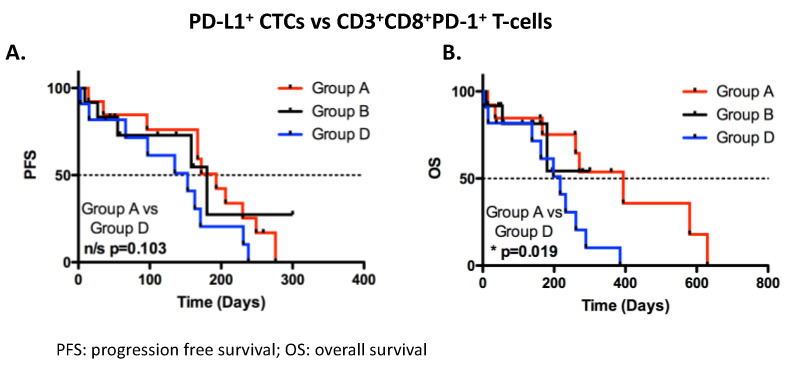
Prognostic significance of PD-L1^+^ CTCs and CD3+CD8^+^ PD-1^+^ T-cells in patients with ES-SCLC. Kaplan–Meier plots of (**A**) PFS and (**B**) OS in patients separated into groups, as follows: group A—high CD3^+^CD8^+^PD-1^+^ T-cells (roc cutoff) with PD-L1^+^CTCs (n = 13); group B—high CD3^+^CD8^+^PD-1^+^ T-cells (roc cutoff) without PD-L1^+^CTCs (n = 12); and group D—low CD3^+^CD8^+^PD-1^+^ T-cells (roc cutoff) without PD-L1^+^CTCs (n = 11). Group C—low CD3^+^CD8^+^PD-1^+^ T-cells (roc cutoff) with PD-L1^+^CTCs (n = 1) was not plotted as there was only one patient. For PFS: log-rank test group A vs. group D, *p* = 0.103, and for OS: log-rank test group A vs. group D, *p* = 0.019. n/s: non-significant; * *p* < 0.05. Unpaired Student *t* test.

**Table 1 biomedicines-12-00146-t001:** SCLC patients’ clinical characteristics.

Characteristics	Sub-Categories	Values
Median age		70 years (range 44–84 years)
Gender	Male	34 (79%)
	Female	9 (21%)
Stage	Limited	6 (14%)
	Extensive	37 (86%)
Metastasis	Brain	11 (25%)
	Lung	13 (30%)
	Liver	15 (35%)
	Bones	8 (17%)
	Adrenal gland	6 (14%)
	LNs	7 (16%)
	Pleural	14 (33%)
	other	4 (9%)
Best Response	PR	26 (60%)
	SD	1 (2%)
	Mixed	2 (5%)
	Unknown	14 (33%)
Smoking Status	Never	1 (2%)
	Former	10 (23%)
	Curent	28 (65%)
	Unknown	4 (9%)
	<40 pack year	8 (19%)
	40–80 pack year	13 (31%)
	>80 pack year	15 (36%)
	Unknown	6 (14%)

**Table 2 biomedicines-12-00146-t002:** Correlations between CTCs and different T-cell populations. In red statistically significant correlations.

		PD-1^+^ CD3^+^CD8^+^ T-Cells			PD-1^+^ CD3^+^CD4^+^ T-Cells		
CTCs		Total	Teff	Tem	Total	Teff	Tem
**CK+CD45-**	Spewrman r	0.351	−0.94	0.011	0.010	−0.137	−0.188
	*p*-value	** 0.038 **	0.611	0.952	0.957	0.485	0.336
	95% CI	0.010 to 0.619	−0.443 to 0.278	−0.354 to 0.373	−0.374 to 0.392	−0.494 to 0.259	−0.533 to 0.209
**CK+CD45-PD-L1-**	Spewrman r	−0.064	−0.165	−0.72	−0.059	−0.198	−0.276
	*p*-value	0.713	0.374	0.697	0.765	0.310	0.154
	95% CI	−0.398 to 0.284	−0.499 to 0.211	−0.425 to 0.299	−0.433 to 0.330	−0.540 to 0.199	−0.596 to 0.119
**CK+CD45-PD-L1+**	Spewrman r	0.342	−0.132	−0.65	0.064	−0.132	−0.119
	*p*-value	** 0.044 **	0.477	0.727	0.744	0.501	0.544
	95% CI	−0.000 to 0.612	−0.473 to 0.243	−0.419 to 0.306	−0.326 to 0.436	−0.490 to 0.264	−0.480 to 0.276

CTC: circulating tumor cell; CK: Cytokeratin; PD-1: programmed cell death receptor 1; PD-L1: programmed death ligand 1; Teff: CD45RA^+^CD45RO^-^CCR7^-^; Tem: CD45RA^-^CD45RO^+^CCR7^-^.

**Table 3 biomedicines-12-00146-t003:** Associations between different T-cell phenotypes and clinical outcomes of treatment-naïve ES-SCLC patients. In red statistically significant correlations.

			Progresion Free Survival			Overall Survival		
T-Cell Populations	ROC Cut Off		Median (Days)	95% HR CI	*p* Value	Median (Days)	95% CI	*p* Value
**CD3^+^CD8^+^**	24	High	171	0.1988 to 1.283	0.150	271	0.1498 to 1.317	0.143
		Low	153			198		
**CD3^+^CD8^+^PD-1**	22	High	172	0.1731 to 0.9486	** 0.037 **	394	0.1830 to 1.054	0.065
		Low	153			217		
**CD8^+^ Teff**	25	High	180	0.1402 to 0.8555	** 0.021 **	261	0.1826 to 1.273	0.140
		Low	158			232		
**CD8^+^ Teff PD-1**	12.7	High	167	0.1562 to 1.450	0.191	289	0.03718 to 0.6173	** 0.008 **
		Low	149			190		
**CD8^+^ Tem**	33.6	High	164	0.8099 to 3.692	0.157	232	0.7701 to 5.062	0.156
		Low	167			394		
**CD8^+^ Tem PD-1**	38	High	172	0.4013 to 2.091	0.835	394	0.2135 to 1.424	0.218
		Low	164			260		
**CD3^+^CD4^+^**	41	High	158	0.4671 to 2.096	0.977	385	0.2410 to 1.380	0.216
		Low	165			246		
**CD3^+^CD4^+^PD-1**	7.5	High	158	0.3293 to 1.667	0.468	289	0.2277 to 1.453	0.242
		Low	164			232		
**CD4^+^ Teff**	10.7	High	157	0.4969 to 3.160	0.632	261	0.2563 to 1.668	0.374
		Low	164			232		
**CD4^+^ Teff PD-1**	18	High	163	0.6510 to 3.038	0.385	232	0.6423 to 3.869	0.320
		Low	167			271		
**CD4^+^ Tem**	33.2	High	167	0.2170 to 1.573	0.287	232	0.4638 to 3.653	0.616
		Low	153			337		
**CD4^+^ Tem PD-1**	7.7	High	158	0.5744 to 2.720	0.573	232	0.5109 to 3.050	0.626
		Low	186			265		

PD-1: programmed cell death receptor 1; Teff: CD45RA^+^CD45RO^-^CCR7^-^; Tem: CD45RA^-^CD45RO^+^CCR7^-^; ROC: receiver operating characteristic; HR: hazard ratio, CI: confidence interval estimate.

## Data Availability

Data presented in the study are available upon request from the corresponding author.

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
