# Peer review of "ES-SCLC Patients with PD-L1^+^ CTCs and High Percentages of CD8^+^PD-1^+^T Cells in Circulation Benefit from Front-Line Immunotherapy Treatment"

_biomedicines, 2024, doi:10.3390/biomedicines12010146_

Round 1

Reviewer 1 Report

Comments and Suggestions for Authors

Major comments:

1.     There are 18 CK+CD45- patients. But why 14 of them exhibit PD-L1-positive and 10 of them exhibit PD-L1-negative (Fig 3B). The total number is 24, which is not consistent?

2.     The description in the Results is not matched to the results in Fig 3C. 42% is PD-L1-positive and 58% is PD-L1-negative, but Fig 3C reveals another story.

3.     It may be Group A vs Group D in Fig 4 according to Table S2. Please check.

Minor comments:

1.     Please notice that there are question markers in the figures.

Author Response

We appreciate the time and efforts by the editor and referees in reviewing this manuscript. 

The manuscript has been revised on a point by point basis addressing reviewer comments. All the changes are also highlighted in yellow in the revised manuscript. We believe that the revised version can meet the journal publication requirements.

Response to Comments from Reviewer 1

Major concers:

Comment 1: There are 18 CK+CD45- patients. But why 14 of them exhibit PD-L1-positive and 10 of them exhibit PD-L1-negative (Fig 3B). The total number is 24, which is not consistent?

Response 1:

We appreciate your insightful comment. Indeed, there are 18 CK+CD45- patients, 14 of them exhibit PD-L1-positive and 10 of them exhibit PD-L1-negative CTCs (Fig 3B). The total number is 18 because 7 patients have both PD-L1-positive and PD-L1-negative CTCs. To clarify this issue in the revised version of this manuscript Table S1 was included where the numbers of  CK+CD45- , CK+CD45-PD-L1+ and CK+CD45-PD-L1- for each patient are included.

Comment 2: The description in the Results is not matched to the results in Fig 3C. 42% is PD-L1-positive and 58% is PD-L1-negative, but Fig 3C reveals another story.

Response 2:

Thank you for this valuable suggestion. In the revised version of this manuscript, we corrected this issue and also we have included a new version of Fig 3C.

Comment 3: It may be Group A vs Group D in Fig 4 according to Table S2. Please check.

Response 3:

Thank you for this valuable suggestion. In the revised version of this manuscript, we have included a new version of Fig 4 where Group A is compared with Group D.

minor concerns:

Comment 1: Please notice that there are question markers in the figures.

Response 1: Thank you for this helpful suggestion. In the revised version of this manuscript the question marks have been deleted.

We are grateful to the reviewers for their critical contribution and crucial comments. We hope that we have succeeded in addressing all the issues raised by the reviewers and that you find our research suitable for publications in the Biomedicines Journal.

Reviewer 2 Report

Comments and Suggestions for Authors

Please clarify the exclusion and inclusion creteria and the type of the study

please, clarify the size of subgroups with different disease extent

Which version of RECIST criteria was applied? What instrumental test was used for staging?

it should be clarified which groups the log rank test was applied to

please add more information: how the tissues were obtained and then patient information such as comorbidities and pack-year

Why didn't you use cox-regression analyisis to correlate some parameters to the survival?

I suggest to include a reference in the discussion to highlight the different fields of apllication of circulating tumor cells

Transl Lung Cancer Res. 2021 Jan;10(1):80-92.

Author Response

We appreciate the time and efforts by the editor and referees in reviewing this manuscript. 

The manuscript has been revised on a point by point basis addressing reviewer comments. All the changes are also highlighted in yellow in the revised manuscript. We believe that the revised version can meet the journal publication requirements.

Response to Comments from Reviewer 2

Comment 1: Please clarify the exclusion and inclusion creteria and the type of the study

Response 1: Thank you for the important comment. In the revised version of this manuscript, we have included the type of the study as well as the inclusion and exclusion criteria. Please check the paragraph 2.1. Patients and Blood collection at the Materials and Methods section.

Comment 2: Please, clarify the size of subgroups with different disease extent

Response 2: Thank you for this suggestion. In the revised version of this manuscript, we have included the number as well as the percentage of patients with Extensive and Limited stage disease. Please check the paragraph 2.1. Patients and Blood collection at the Materials and Methods section.  Additionally, you can also find this information in the Table 1.

Comment 3: Which version of RECIST criteria was applied? What instrumental test was used for staging?

Response 3: Thank you for this valuable suggestion. In the present study the Response Evaluation Criteria in Solid Tumors (RESIST) (version 1.1). This was included in the revised version of this manuscript. Please check the paragraph 2.1. Patients and Blood collection at the Materials and Methods section.

Moreover, for staging, patients CT scans as well as MRIs at the time of diagnosis were used.  

Comment 4: It should be clarified which groups the log rank test was applied to

Response 4: Thank you for this helpful suggestion. In the revised version of this manuscript, in Fig.4 as well as in Fig.4 legent we have included the the groups that log rank test was applied.

Comment 5: Please add more information: how the tissues were obtained and then patient information such as comorbidities and pack-year

Response 5:  Thank you for this suggestion. In the revised version of this manuscript, you can find a new version of Table 1 were more information about patients was included such as smoking status and pack year.

Moreover, for histologically confirmation of SCLC at diagnosis patients’ tissue was obtained by bronchoscopy or EBUS (endobronchial ultrasound) bronchoscopy. This information is included in the revised form of this manuscript. Please check the paragraph 2.1. Patients and Blood collection at the Materials and Methods section.

Comment 6: Why didn't you use cox-regression analyisis to correlate some parameters to the survival?

Response 6: Thank you for the comment. In the revised version of this manuscript a cox regression analysis has been included testing different clinical parameters. You can find this analysis as Supl. Table 4.

Comment 7: I suggest to include a reference in the discussion to highlight the different fields of apllication of circulating tumor cells

Transl Lung Cancer Res. 2021 Jan;10(1):80-92.

Response 7: Thank you for the suggestion. In the revised version of this manuscript a paragraph describing the different applications of CTCs has been included in the discussion.

We are grateful to the reviewers for their critical contribution and crucial comments. We hope that we have succeeded in addressing all the issues raised by the reviewers and that you find our research suitable for publications in the Biomedicines Journal.

Reviewer 3 Report

Comments and Suggestions for Authors

Dr. Xagara et al. in this original article summarizes how ES-SCLC patients with high PD-L1 expressing CTC and high PD-1 expressing CD8 cells benefit from immunotherapy in front line.

This original article is well explained but I have few comments which need to be addressed-

1-    Please correct in title- frond line or front-line in the title.

2-    Please include a section with proper reference about the four different subtypes of SCLC and where immunotherapy works best in introduction.

3-    From figure 1- % of PD1+ CD8+ T cells is significantly high in ED. But my first concern is about the sample size- HD- 10, LD-6 and ED-37. Please include a decent number of healthy CD8+ T cells in the experiment to conclude.

4-    In figure-1 with ED PD1+ CD8+ T cells are distributed in two/ three segment. Is it possible to understand if the upper 7 patients are withing SCLC-I (mean do not express all three genes-ASCL1,NEUROD1,and POU2F3 from other subtypes.

5-    Figure 2 A- Please include X and Y axis details as well as the MFI in the flow histogram.

6-    Can author please provide examples of CTC stained slide images in the manuscript.

Author Response

We appreciate the time and efforts by the editor and referees in reviewing this manuscript. 

The manuscript has been revised on a point by point basis addressing reviewer comments. All the changes are also highlighted in yellow in the revised manuscript. We believe that the revised version can meet the journal publication requirements.

Response to Comments from Reviewer 3

Comment 1: Please correct in title- frond line or front-line in the title.

Response 1: We appreciate your insightful comment. The title has been corrected.

Comment 2: Please include a section with proper reference about the four different subtypes of SCLC and where immunotherapy works best in introduction.

Response 2:Thank you for this valuable suggestion. In the revised version of this manuscript, we included a paragraph describing the four different molecular subtypes of SCLC.

Comment 3: From figure 1- % of PD1+ CD8+ T cells is significantly high in ED. But my first concern is about the sample size- HD- 10, LD-6 and ED-37. Please include a decent number of healthy CD8+ T cells in the experiment to conclude.

Response 3: Thank you for this suggestion. In the revised version of this manuscript, we have included a new version of Fig 1 where 10 additional healthy donors (20 in total) were analyzed for different sub-populations of CD8+ T-cells.

Comment 4:  In figure-1 with ED PD1+ CD8+ T cells are distributed in two/ three segment. Is it possible to understand if the upper 7 patients are withing SCLC-I (mean do not express all three genes-ASCL1,NEUROD1,and POU2F3 from other subtypes.

Response 4: Thank you for this helpful suggestion. Unfortunately, we have not available primary tissues from the patients of this study so we cannot check the mRNA levels of the mentioned genes in order to categorize them.

Comment 5:  Figure 2 A- Please include X and Y axis details as well as the MFI in the flow histogram.

Response 5: Thank you for this suggestion. In the revised version of this manuscript the axis details as well as MFI were included in figure 2A.

Comment 6:  Can author please provide examples of CTC stained slide images in the manuscript.

Response 6: Thank you for this valuable suggestion. In the revised version of this manuscript we have included images of stained CTCs as supl Fig.1.

We are grateful to the reviewers for their critical contribution and crucial comments. We hope that we have succeeded in addressing all the issues raised by the reviewers and that you find our research suitable for publications in the Biomedicines Journal.

Round 2

Reviewer 1 Report

Comments and Suggestions for Authors

No more questions.